# A normative modelling approach reveals age-atypical cortical thickness in a subgroup of males with autism spectrum disorder

Richard A. I. Bethlehem [1,2✉], Jakob Seidlitz[1,3], Rafael Romero-Garcia[1], Stavros Trakoshis [4,5], Guillaume Dumas [6,7,8] & Michael V. Lombardo [2,4]

Understanding heterogeneity is an important goal on the path to precision medicine for autism spectrum disorders (ASD). We examined how cortical thickness (CT) in ASD can be parameterized as an individualized metric of atypicality relative to typically-developing (TD) age-related norms. Across a large sample ($n = 870$ per group) and wide age range (5–40 years), we applied normative modelling resulting in individualized whole-brain maps of age-related CT atypicality in ASD and isolating a small subgroup with highly age-atypical CT. Age-normed CT scores also highlights on-average differentiation, and associations with behavioural symptomatology that is separate from insights gleaned from traditional case-control approaches. This work showcases an individualized approach for understanding ASD heterogeneity that could potentially further prioritize work on a subset of individuals with cortical pathophysiology represented in age-related CT atypicality. Only a small subset of ASD individuals are actually highly atypical relative to age-norms. driving small on-average case-control differences.

[1] Brain Mapping Unit, Department of Psychiatry, University of Cambridge, Cambridge CB2 0SZ, UK. [2] Autism Research Centre, Department of Psychiatry, University of Cambridge, Cambridge CB2 8AH, UK. [3] Department of Child and Adolescent Psychiatry and Behavioral Science, Children's Hospital of Philadelphia, Philadelphia, PA 19104, USA. [4] Laboratory for Autism and Neurodevelopmental Disorders, Center for Neuroscience and Cognitive Systems @UniTn, Istituto Italiano di Tecnologia, Rovereto, Italy. [5] Department of Psychology, University of Cyprus, Nicosia, Cyprus. [6] Human Genetics and Cognitive Functions Unit, Institut Pasteur, Paris, France. [7] CNRS UMR3571 Genes, Synapses and Cognition, Institut Pasteur, Paris, France. [8] Human Genetics and Cognitive Functions Unit, University Paris Diderot, Sorbonne Paris Cité, Paris, France. ✉email: rb643@medschl.cam.ac.uk

Autism spectrum disorder (ASD) is a clinical behavioural consensus label we give to a diverse collection of patients with social-communication difficulties and pronounced repetitive, restricted, and stereotyped behaviours[1]. Beyond the single label of ASD, patients are in fact widely heterogeneous in phenotype, but also with regards to the diversity of different aetiologies[2]. Even within mesoscopic levels of analysis such as examining brain endophenotypes, heterogeneity is the rule rather than the exception[3]. At the level of structural brain variation, neuroimaging studies have identified various neuroanatomical features that might help identify individuals with autism or reveal elements of a common underlying biology[3]. However, the vast neuroimaging literature is also inconsistent, with reports of hypo- or hyper-connectivity, cortical thinning versus increased grey or white matter, brain overgrowth, arrested growth, or even lack of morphological difference altogether, etc.[4–13], leaving stunted progress towards understanding mechanisms driving cortical pathophysiology in ASD and translating neuroimaging into clinical utility.

Multiple explanations could be behind this inconsistency across the literature. Methodology widely differs across studies (e.g., low statistical power, different ways of estimating morphology or volume) and is likely a very important factor[9,14]. Initiatives such as the autism brain imaging data exchange (ABIDE[15]); have made it possible to boost sample size by pooling together data from several different studies. However, within-group heterogeneity in the autism population also immediately stands out as another factor obscuring consistency in the literature, especially when the dominant approach of case-control models largely ignores heterogeneity within the ASD population. In particular, some autism-related heterogeneity reported in literature might be explained by factors such as age[16,17]. Indeed, with regards to structural brain features of interest for study in ASD (e.g., volume, cortical thickness (CT), surface area), these features change markedly over development and may follow altogether different trajectories in ASD[18–20]. Typical approaches towards dealing with age revolve around group statistical modelling of age as the variable of interest or removing age as a covariate and then parametrically modelling on-average differences between cases versus controls. While these are common approaches in the literature, they do not immediately provide individualized estimates of age-related atypicality nor do they account for individual variation in developmental trajectories. In contrast, normative models of age-related variation may likely be an important alternative to these approaches and may mesh better with some conceptual views of atypicality in ASD as being an extreme of typical population norms[21]. In contrast to the canonical case-control model, normative age modelling allows for computation of individualized metrics that can hone in on the precision information we are interested in—that is, atypicality of development expressed in specific ASD individuals relative to non-ASD norms. Such an approach may be a fruitful way forward in isolating individuals who are 'statistical outliers'. The reasons behind why these individuals are outliers relative to non-ASD norms may be of potential clinical and/or mechanistic importance. Furthermore, conventional case-control analyses may obscure more subtle individual differences as they assume on-average group differences. This is especially important in light of previously reported null-findings[14]. Indeed, if we are to move forward towards stratified psychiatry and precision medicine for ASD[22], we must go beyond case-control approaches and employ dimensional approaches that can tell us information about which individuals are atypical and how or why they express such atypicality. Thus, this approach aims to provide more than a mere statistical advance, it aims to better conceptualize and capture personalized inferences that may ultimately result in more meaningful and targeted clinical inference.

In the present study, we employ normative modelling on age-related variability as a means to individualize our approach to isolate specific subsets of patients with very different neural features. Here we focus specifically on a neural feature of cortical morphology known as CT. CT is a well-studied neuroanatomical feature thought to be differentially affected in autism and has received increasing attention in recent years[23–27]. Recent work from our group also identified a genetic correlate for autism-specific CT variation despite considerable heterogeneity in group-specific CT in children with autism[28]. A study examining ABIDE I cohort data discovered case-control differences in CT, albeit very small in effect size[14]. Similarly, the most recent and largest study to date, a mega-analysis combining data from ABIDE and the ENIGMA consortium, also indicated very small on-average case-control differences in CT restricted predominantly to areas of frontal and temporal cortices, and indicate very subtle age-related between-group differences and substantial within-group age-related variability[27]. Another recent study also highlighted widespread differences in CT and several differences that were sex-specific[7]. Overall, these studies emphasize three general points of importance. First, age or developmental trajectory is extremely important[16,29–31]. Second, given the considerable within-group age-related variability and the presence of a large majority of null and/or very small between-group effects, rather than attempting to find on-average differences between all cases versus all controls, we should shift our focus to capitalize on this dimension of large age-related variability and isolate autism cases that are at the extremes of this dimension of normative variability. Third, biological sex is likely to be an important modulator of ASD-specific morphological differences[7].

Given our approach of age-related normative CT modelling, we first compare the utility of age-related normative modelling directly against more traditional case-control models. We then describe the prevalence of ASD cases that show meaningful age-related deviance in CT (i.e. >2 standard deviations from age-related norms or outside the 95% population confidence bounds) and show how a metric of continuous variability in age-related atypicality in CT is expressed across the cortex in autism. Finally, we explore age–atypical CT–behaviour associations and assess whether such dimensional analyses associated with behaviour identify similar or different regions than typical case-control analyses. To show applicability of this approach we also applied the same method to other measures of neuroanatomy; gyrification, volume and surface area. Results and analyses of these metrics can be found in the supplementary materials and all code and data used are available on GitHub[32].

## Results

**Age-related normative modelling**. Normative modelling of age-related CT effects was done utilizing male-only data from the typically developing group (TD) (see "Methods" section for full sample description, Fig. 1 for a schematic overview and Supplementary Figs. 1 and 2 for more demographics information). All analyses were done on CT averaged within 308 cortical regions[33]. We used a local polynomial regression fitting procedure (LOESS[34,35], where the local width or smoothing kernel of the regression was determined by the model that provided the overall smallest sum of squared errors using hyperparameter optimization across 5–100% of the full age range using Brent's method[36] as implemented in the R optim function from the stats package. We also assessed consistency of our output using centiles scoring and consistency of the normative model using extensive bootstrapping and sensitivity analyses, both showed high outcome consistency (see "Methods" section and Supplementary Materials; Supplementary Figs. 3–5). To align the TD and ASD groups,

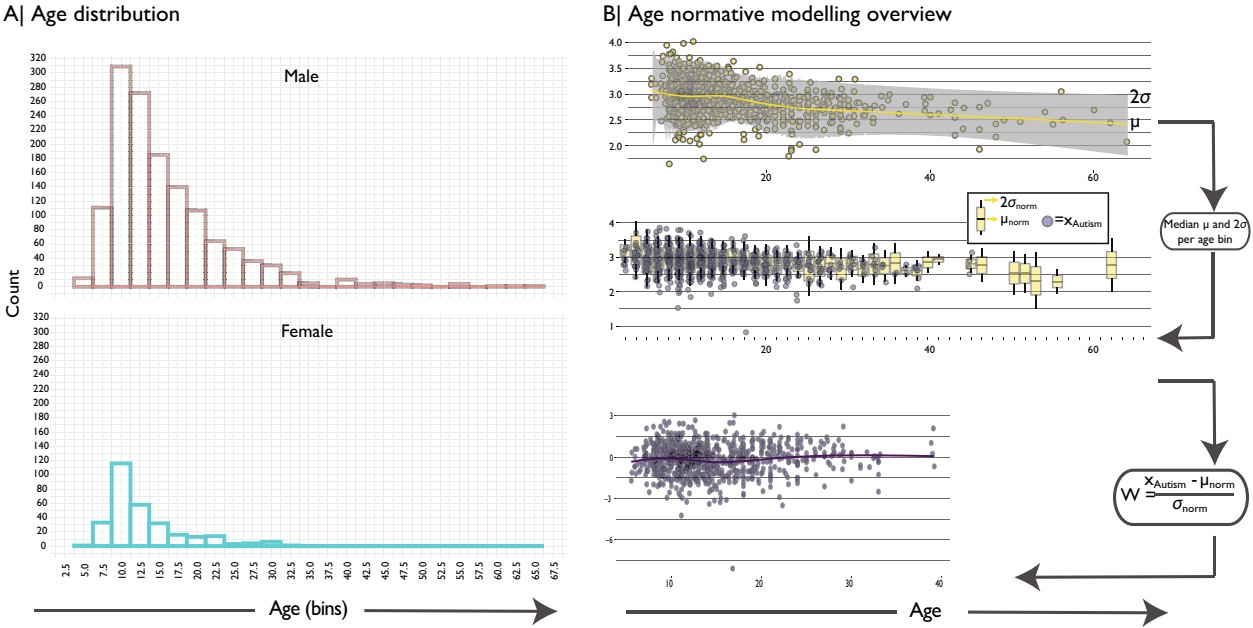

**Fig. 1 Demographics and descriptive statistics. a** Histogram of age distribution per sex. Females were excluded from further analyses due to known sex differential effects in autism and the lack of available data to estimate population norms (see "Methods" section for details). **b** Schematic overview of normative modelling. In the first instance LOESS regression is used to estimate the developmental trajectory on CT for every individual brain region to obtain an age-specific mean and standard deviation. Then we computed median for each one-year age-bin for these mean and median neurotypical estimates to align them with the ASD group. Next, for each individual with autism and each brain region the normative mean and standard deviation are used to compute a w-score relative to their neurotypical age-bin. Contrary to conventional boxplots, the second panel shows mean, 1 sd and 2 sd for the neurotypical group (in yellow) and individuals with an autism diagnosis in purple.

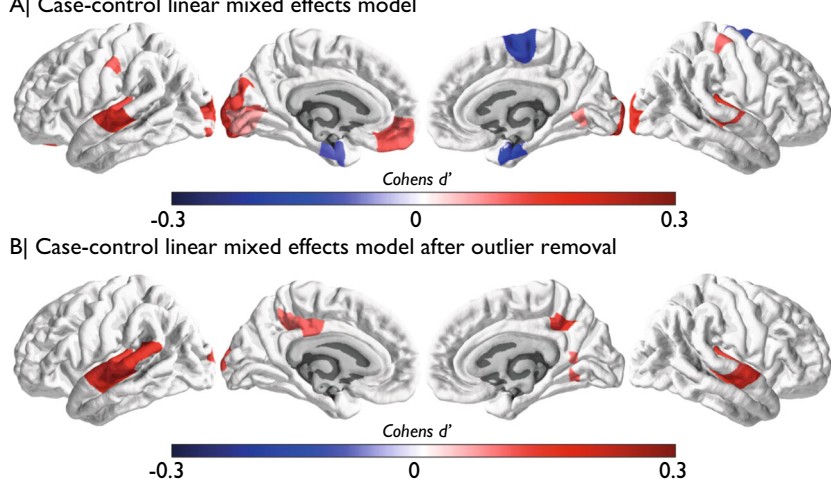

**Fig. 2 Case control difference analysis with linear mixed effect model.** Panel **a** shows effect sizes for regions passing FDR correction for linear mixed effect modelling of conventional case control difference analysis. Cohen's *d* values represent ASD−control, thus blue denotes ASD<control and red denotes ASD>control. Panel **b** shows effect sizes for regions passing FDR correction after outlier removal for the same linear mixed effect modelling of conventional case control difference analysis.

both were binned into one-year age bins. For each age bin and every brain region we computed a normative mean and standard deviation from the TD group. This was done separately for each sex, given known sex differential developmental trajectories. These statistical norms were then used to compute a w-score (analogous to a z-score) for every individual with autism and every brain region as follows:

$$W_{\text{region}} = \frac{\text{CT}_{\text{region}} - \mu_{\text{norm}_{\text{region}}}}{\sigma_{\text{norm}_{\text{region}}}}$$

The w-score for an individual thus reflects how far away their CT is from TD norms in units of standard deviation. Because w-scores are computed for every brain region, we get a w-score map for each ASD participant showing how each brain region for that individual is atypical relative to TD norms. Age bins that contained fewer than five data-points in the TD group were excluded from subsequent analysis as the standard deviations for these bins would essentially be zero (and thus the w-score could not be computed). With the inclusion of motion we also excluded individuals for which no resting-state fMRI was available. The characteristics of the final autism sample are listed in Table 1.

**Case-control differences versus age-normative modelling.** Our first analysis examined conventional case-control differences using linear mixed effect modelling including site, sex, age, in-scanner head motion[37] and Euler index[38] as covariates. As expected from prior papers utilizing large-scale datasets for case-control analysis[14,27], a small subset of regions (8.7%, 27/308 regions) pass FDR correction. Of these regions, most are of small effect size, with 26 of the detected 27 regions showing an effect <0.2 standard deviations of difference (Fig. 2a). We suspected that such small effects could be largely driven by a few ASD patients[39] with highly age-atypical CT. Because we also had computed w-scores from our normative age-modelling approach, we identified specific 'statistical outlier' patients for each individual region with w-scores >2 standard deviations from typical norms and excluded them from the case-control analysis. This analysis guards against the influence of these extreme outliers, and if there are true on-average differences in ASD, the removal of these outlier patient*regions should have little effect on our ability to detect case-control differences. However, removal of outlier patients now revealed only 14 significant regions instead of 27 regions with small case-control differences—a 1.9-fold decrease in the number of regions detected. Indeed, the majority of case-control differences identifying small on-average effects were primarily driven by this small subset of highly atypical patients (Fig. 2b). These remaining 14 regions with small on-average effects were restricted to areas near the posterior cingulate cortex, temporo-parietal cortex and areas of visual cortex.

In contrast to a canonical case-control model, we computed normative models of age which resulted in individualized w-scores that indicate how atypical CT is for an individual compared to typical norms for that age. This modelling approach allows for computation of w-scores for every region and in every patient, thus resulting in a w-score map that can then itself be tested for differences from a null hypothesis of w-score = 0,

indicating no significant on-average ASD atypicality in age-normed CT. These hypothesis tests on normative w-score maps revealed no regions surviving FDR correction.

**Isolating ASD individuals with age-related CT atypicality.** While the normative modelling approach can be sensitive to different pathology than traditional case-control models, another strength of the approach is the ability to isolate individuals expressing high CT-atypicality. We operationalized 'significant' atypicality in statistical terms as w-scores >2 SD away from TD norms. By applying this cut-off, we can then describe what proportion of the ASD population falls into this CT subgroup category for each individual brain region. Over all brain regions the median prevalence of these patients is around 7.6% (Fig. 3). Meaning that in each brain region there are ~7.6% of individuals that would be considered an outlier. This difference from an expected proportion of 5% in the present sample corresponds to a $X^2$ of 3.85 (with Yates continuity correction[40]) that is significant at $p = 0.049$ (without continuity correction: $X^2 = 4.32$, $p = 0.038$). The distribution of prevalence across brain regions also has a positive tail indicating that for a small number of brain regions the prevalence can jump up to more than 10%. Expressed back into sample size numbers, if 10% of the ASD population had significant CT abnormalities, with a sample size of $n = 699$, this means that $n = 70$ patients possess such atypicality. Underscoring the prevalence of these cases is important since as shown earlier, it is likely that primarily these 'statistical outlier' patients drive most of the tiny case-control differences observed.

There are other interesting attributes about this subset of brain regions. With regard to age, these patients were almost always in the age range of 6–20, and were much less prevalent beyond age 20 (Supplementary Figs. 6 and 7). The median age of outliers across brain regions ranged from [10.6–20.2] years old, with an overall skewed distribution towards the younger end of the spectrum (Supplementary Fig. 7), showing that CT atypicality potentially normalizes with increasing age in ASD, though it should be noted that this may partially be explained by the overall skewed age distribution in the overall dataset.

Patients with CT atypicality were also largely those that expressed such atypicality within specific brain regions and were not primarily subjects with globally atypical CT. To show this we computed a w-score ratio across brain regions (Supplementary

**Table 1 Sample age characteristics after normative modelling selection.**

| Dx | Mean | SD | N | Median | Min | Max |
|---|---|---|---|---|---|---|
| Autism | 14.93 | 5.97 | 699 | 13.40 | 5.53 | 39.2 |
| TD | 15.35 | 6.37 | 624 | 13.34 | 5.89 | 39.4 |

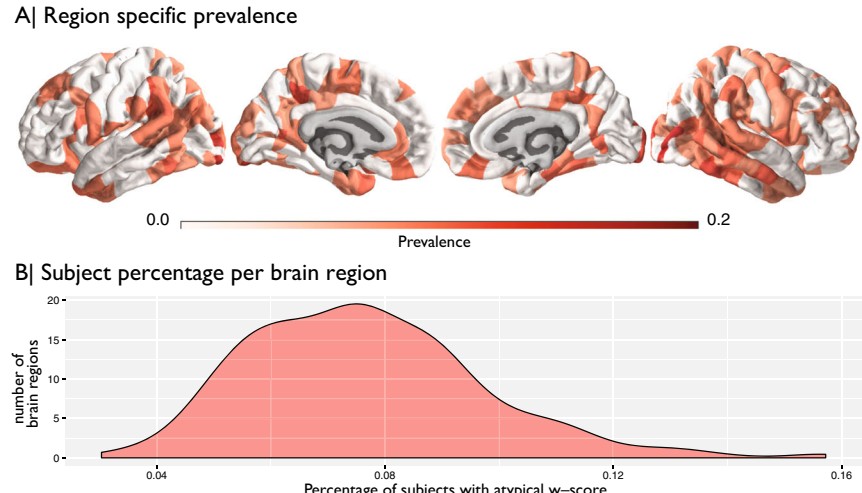

A| Region specific prevalence

0.0                                              0.2
Prevalence

B| Subject percentage per brain region

**Fig. 3 Region specific prevalence of atypical w-scores.** Panel **a** shows the by region prevalence of individuals with a w-score of greater than ±2SD. For visualization purposes these images are thresholded at the median prevalence of 0.076. Panel **b** shows the overall distribution of prevalence across all brain regions.

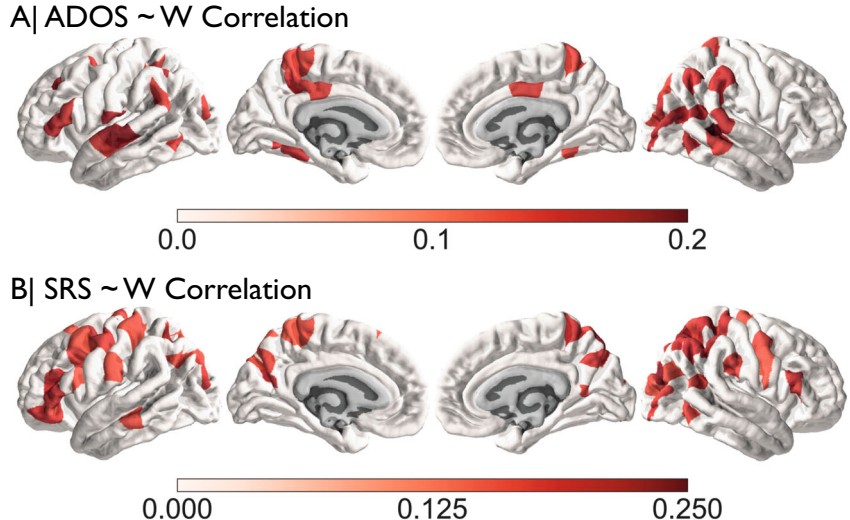

**Fig. 4 Phenotype–w-score correlations.** Spearman correlations between ADOS and w-score in the top panel. The lower panel shows the same for the SRS.

Figs. 6 and 7) that helps us isolate patients that show globally atypical CT across most brain regions. The small number of patients with a ratio indicating a global difference (ratio > 0.5, $n = 14$) were those that had globally thinner cortices. This small subset of individuals was much smaller than the number of region-wise outliers as shown in Fig. 3. Upon visual inspection of the raw data for these participants, it is clear that the global thinning effect is not likely a true biological difference but rather one likely driven by the quality of the raw images, even though the Euler index did not indicate failure in reconstruction. Unfortunately, we did not have enough complete phenotypic data on these subjects to warrant further in-depth phenotypic analysis.

**Exploratory analysis of brain–behaviour relationships**. An additional advantage of the use of normative modelling over the traditional case-control modelling is that we can use the individualized atypicality as a novel metric for finding associations with phenotypic features. Here we used w-scores to compute Spearman correlations for the most commonly shared phenotypic features in the ABIDE dataset: ADOS, SRS, SCQ, AQ, FIQ and Age. After correcting for multiple comparisons across phenotype and region (6 phenotypic measures*308 regions = 1848 tests) we identified a number of brain regions that survive multiple comparison corrections for the SRS and ADOS scores (Fig. 4, Supplementary Fig. 8). SRS is associated with w-scores primarily in areas of lateral frontal and parietal cortex, while ADOS is associated with w-scores primarily in lateral and inferior temporal cortex. Notably, these regions are largely different from regions that appear to show on-average differentiation in case-control and w-score analyses.

**Sensitivity analysis**. Sensitivity analyses on the effects of reconstruction quality using Euler index as well as residual effects of in-scanner head motion from the resting-state acquisition did not reveal a significant impact on thresholded case-control differences or w-score. Specifically, systematic exclusion of top motion and Euler subjects resulted in highly spatially consistent effect size maps (all $r < 0.7$). Individuals identified as statistical outliers did not have disproportionally high motion or high Euler indices. For more details see "Methods" section and supplementary materials (Supplementary Figs. 2 and 3).

**Discussion**

In the present study, we find that with a highly powered dataset, conventional case-control analyses reveal small differences in CT in autism and are restricted to a small subset of regions. In general, this idea about subtle effect sizes for case-control comparisons is compatible with other recent papers utilizing partially overlapping data—Haar and colleagues utilized only ABIDE I data[14], while van Rooji and colleagues[27] utilized both ABIDE I and II dataset combined with further data from the ENIGMA consortium. While these statements about small effect sizes are not novel, our findings suggest that even these small effect sizes may be misleading and over-optimistic. Utilizing normative modelling as a way of identifying and removing CT-atypical outlier patients, we find here that most small case-control differences are driven by a small subgroup of patients with high CT-atypicality for their age, which indeed begs the question of the existence of on-average atypical cortical morphology in autism[14]. In contrast, we further showed that analysis of CT-normed scores (i.e. w-scores) themselves reveals a completely different set of regions that are on-average atypical in ASD. The directionality of such differences also reverses in some cases. For instance, Haar and colleagues discovered that areas of the visual cortex are thicker in ASD compared to TD in ABIDE I[14]. Our case-control analyses here largely mirror that finding. However, re-analysis after w-score outlier removal totally removes the effects previously reported in the visual cortex. Thus, here is a clear case whereby our normative age modelling approach identifies effects that are likely driven by only a small subset of individuals. New insights via normative age modelling, alongside cleaning up interpretations behind case-control models, both highlight the meaningful utility of this approach. The presence of small region-dependent outlier effects in ASD misleadingly drives on-average inferences from case-control models. Thus, it is important for the field to better understand how prevalent this atypicality is for a given brain region (i.e. our analyses did not reveal a consistent brain region or group of individuals with spatially overlapping patterns of extreme w-scores).

We also noted that this region-specific small subgroup showing highly age-atypical CT was predominantly restricted to the childhood to early adult age range. In later adult ages, the prevalence of this subgroup drops off. This could be a potential indicator that highly atypical CT is more prevalent and detectable at earlier ages. It will be important to assess even earlier age

ranges such as the first years of life[30], as well as later adult years when aging processes begin to take effect[41]. Again, it should be noted that this may partially be explained by the overall skewed age distribution in the overall dataset. Future studies with either more balanced developmental samples or samples that cover the entire lifespan will be better positioned to confirm this age-related skewed profile in 'atypical' brain regions.

In addition, we also identify a very small group of individuals that have atypical patterns in over 50% of brain regions. Unfortunately, not much behavioural or phenotypic information was available for this sub-group. We hope that future studies will obtain more detailed phenotypic information in order to delineate more precisely what the clinical and or more broad behavioural implications might be of this atypicality. Furthermore, it is clear from the present work that this subgroup only covers a very small subset in the autism population and thus future studies will require large sample sizes to be able to identify this subgroup. However, mirroring work in autism genetics, whereby discoveries are continually being made regarding very small proportions of the ASD population being explained by highly penetrant genetic mechanisms[42], it also may be the case that such individuals with highly age-atypical CT are individuals with specific highly penetrant biological mechanisms underlying them, and possibly related to neurogenesis and other factors that are implicated in CT changes[28]. With animal models of highly penetrant genetic mechanisms linked to autism, it is notable that such mechanisms have heterogeneous effects on brain volume[43]. Thus, the fact that this is only a small subset need not be an obstacle for the discovery of core biological mechanisms. Ideally, future studies will also collect detailed genetic and/or other biological information in order to probe the core biological aetiology underlying the pattern of broad atypical CT.

We also conducted exploratory analyses to relate the w-scores back to phenotypic information more broadly, insofar as this was available in ABIDE. Here, we find collections of areas that are largely different from regions normally detected with on-average case-control or on-average non-zero w-score differences. Interestingly, the associations with ADOS and SRS show somewhat differential spatial topography which may suggest that the overall scores are related to different underlying neurobiological mechanisms. Overall, these results could suggest that the normative model is sensitive to signal related to behavioural variability. However, it should be emphasized that ADOS and SRS scores are not available for the full dataset and the reported effects were small and should thus be considered exploratory. In addition, ADOS and SRS are often only collected on individuals with a diagnosis already, which makes the general inference of this brain–behaviour relationship potentially biased. Based on present results however we expect future studies, with more comprehensive phenotypic information such as EU-AIMS2-trials (https://www.aims-2-trials.eu), to be able to confirm this brain–behaviour relationship. It will be interesting to see whether in a larger more comprehensive sample the same topological dissociation becomes apparent as well.

The current results can be contrasted with a recent study on the EU-AIMS LEAP cohort[44]. This study differs from the current work in being based on a completely independent dataset (EU-AIMS LEAP vs. ABIDE). The studies also differ in how normative models are estimated—LOESS and centiles vs. Gaussian process regression. This study applied normative modelling only to males to reduce sex-related heterogeneity whereas Zahibi et al. utilized both males and females and used sex as a factor in the model. The current study also utilizes a larger sample size (autism $n = 699$, TD $n = 624$; autism $n = 321$, TD $n = 206$ in ref. [44]). Despite these differences, some important consistencies emerge. In particular, our map of prevalence of the CT outlier group (Fig. 3) is somewhat consistent with the spatial topology Zahibi and colleagues report for negative deviations from the normative model (e.g., Fig. 4 of ref. [44]). Furthermore, while our analyses of brain–behavioural relationships is limited, there is some consistency across this study and Zahibi et al. with the correlation between ADOS total scores and left inferior frontal gyrus. Thus, despite the methodological differences, the overall consistency suggests that many of the inferences from these works generalize to the autism population.

There are a number of caveats to consider in the present study. First and foremost, the present data are cross-sectional and the normative age modelling approach cannot make claims about trajectories at an individual level. With longitudinal data, this normative modelling approach could be extended. However, at the moment the classifications of highly age-atypical CT individuals are limited to static normative statistics within discrete age-bins rather than based on statistics from robust normative trajectories. The dataset also represents ASD within an age range that misses very early developmental and also very late adulthood periods. Second, the dataset also presents a post-hoc collection of sites accumulated through the ABIDE initiative, whereby scanners, imaging acquisition sequences and parameters, sample ascertainment, etc., are highly heterogeneous. As a result, we observed that site had a large effect on explaining variance in CT and this is compatible with observations made by other studies[14]. Furthermore, it is likely that there may be systematic interactions between scanner site and some variables of interest such as age (e.g. different scanning sites will likely have recruited specific age cohorts). Third, there are a number of different approaches to normative modelling that all have pros and cons (see ref. [45] for an excellent review). We chose to use LOESS estimation as it is computationally efficient and the resulting w-scores are easily interpretable. However, since it is based on estimation of standard deviation from a normative sample it is potentially sensitive to small samples in a given age-bin (e.g. if there are only four data-points for a given age-bin there is likely to be a less reliable sd). Hence in situations where data is spare the LOESS approach may allow for less reliable normative scores. In order to assess the sensitivity of our approach in the present data we implemented the aforementioned bootstrapping procedure to identify robustness of outlier detection. In addition, we also conducted a centiles estimation that is relatively standard in for example epidemiology[46], similar to quantile rank maps[47] and arguably less sensitive to small sample uncertainty. Both approaches showed highly significant correlation in determining whole-brain w-score ratios ($r = 0.87$, $p = 4e{-}119$ and $r = 0.66$, $p = 5.7e{-}39$ for ABIDE I and ABIDE II, respectively; see supplementary materials, Supplementary Fig. 5). Fourth, our current sample was matched on IQ and as a result excluded individuals with low IQ scores ($<70$). While higher IQ does not automatically imply higher overall functioning[48] it does limit the generalisability of our findings to individuals with normal to high IQ. Finally, although in-scanner head motion is a well-known confounder in resting-state connectivity studies[49,50] it has recently been shown that the same motion may also affect structural image quality and surface reconstruction, especially in clinical cohorts[37]. To address this issue in the present analysis we included mean framewise displacement in our models. In this line, Savalia et al. [51] recently showed that framewise displacement is a sensitive proxy of motion-related bias in structural images. We find that, while this severely impacted the conventional case control analysis (e.g. reducing the number of significant ROI's from 38 to 27), it did not impact the outlier thresholded analysis to the same extent. To further assess the sensitivity of motion on the present approach we include sensitivity analyses based on systematic removal of high motion subjects and find that the spatial topology of effects

was strongly conserved. Given the impact on the conventional analysis approach we strongly encourage future studies to consider motion as an important confounder.

In conclusion, the present study shows how normative age modelling approach in ASD questions our interpretation of conventional case-control modelling while shedding new insight into heterogeneity in ASD. We show that results from case-control analyses, even within large datasets, can be highly susceptible to the influence of 'outlier' subjects. Removing these outlier subjects from analyses can considerably clean up the inferences being made about on-average differences that apply to a majority of the ASD population. Rather than only being nuisances for standard group-level analyses, these outlier patients are meaningful in their own light, and can be identified with our normative age modelling approach. Normative models may provide an alternative to case-control models that test hypotheses at a group-level, by allowing additional insight to be made at more individualized levels, and thus help further progress towards personalized medicine for ASD. Furthermore, the current approach is in line with the original normative modelling approach advocated by Marquand and colleagues[21] which suggested the development of methods to move away from the traditional case-vs.-control analyses. Normative modelling was originally proposed as one solution among others like stratification. Here, a clear path forward would be to combine both, for instance by using output of normative models as features used in the participant stratification, thus avoiding trivial clustering caused by confounding factors. In the present work we show that normative modelling is more than a purely statistical advancement to improve robustness. It allows us to identify a small subgroup that we expect to have strong relevance for the discovery of core biological or phenotypic clinical targets. It allowed for exploration of brain–behaviour relationships that reveal differential spatial topology for ADOS and SRS scores. More importantly however, it moves us conceptually closer to making precise dimensional inferences rather than purely relying on diagnostic categories.

## Methods

**Participants.** In this study, we first sought to leverage large neuroimaging datasets to yield greater statistical power for identifying subtle effects. To achieve this, we utilized the ABIDE datasets (ABIDE I and II; 15) (see Supplementary Fig. 1). Informed consent was given at each site included in the ABIDE studies, see the website for more details: http://fcon_1000.projects.nitrc.org/indi/abide/. Given that the normalized modelling approach gives us individual level measures we chose to also include sites with limited numbers of subjects. Groups were subsequently matched on age using the non-parametric nearest neighbour matching procedure implemented in the Matchit package in R (https://cran.r-project.org/web/packages/MatchIt/index.html)[52]. After matching case and control groups and excluding scans of poorer quality (see supplementary materials) we were left with a sample size $N = 870$ per group (Tables 2 and 3).

**Imaging processing and quantification.** Cortical surface reconstruction was performed using the MPRAGE (T1) image of each participant with FreeSurfer (http://surfer.nmr.mgh.harvard.edu/) version (v5.3.0, to ensure comparability with previous ABIDE publications). The reconstruction pipeline performed by FreeSurfer "recon-all" involved intensity normalization, registration to Talairach space, skull stripping, WM segmentation, tessellation of the WM boundary, and automatic correction of topological defects. Briefly, non-uniformity intensity correction algorithms were applied before skull stripping[53], resulting in resampled isotropic images of 1 mm. An initial segmentation of the white matter tissue was performed to generate a tessellated representation of the WM/GM boundary. The resulting surface was deformed outwards to the volume that maximizes the intensity contrast between GM and cerebrospinal fluid, generating the pial surface[54]. Resulting surfaces were constrained to a spherical topology and corrected for geometrical and topological abnormalities. CT of each vertex was defined as the shortest distance between vertices of the GM/WM boundary and the pial surface[55]. We chose to not conduct manual segmentations and excluded failed subjects from any subsequent analysis (and these subjects were removed prior to the matching and QC procedures). To assess the quality of Freesurfer reconstructions we computed the Euler index[38]. The Euler number is a quantitative proxy index of segmentation quality

**Table 2 Sample characteristics of age.**

| Dx | Sex | Mean | SD | N | Median | Min | Max |
| --- | --- | --- | --- | --- | --- | --- | --- |
| Autism | Male | 16.32 | 9.09 | 754 | 13.75 | 5.13 | 64 |
| Autism | Female | 15.06 | 8.43 | 116 | 12.57 | 5.22 | 54 |
| TD | Male | 16.64 | 8.98 | 660 | 13.69 | 5.89 | 64 |
| TD | Female | 13.25 | 5.33 | 210 | 11.09 | 5.91 | 32 |

**Table 3 Sample characteristics.**

| Measure | Dx | Sex | Mean | SD | N | Median |
| --- | --- | --- | --- | --- | --- | --- |
| IQ | Autism | Male | 106.13 | 16.51 | 754 | 107 |
| | Autism | Female | 105.88 | 16.21 | 116 | 106.5 |
| | TD | Male | 111.28 | 12.13 | 660 | 111 |
| | TD | Female | 112.07 | 13.21 | 210 | 112 |
| ADOS | Autism | Male | 11.15 | 3.86 | 505 | 11 |
| | Autism | Female | 11.41 | 3.9 | 63 | 11 |
| | Control | Male | 1.55 | 1.58 | 38 | 1 |
| | Control | Female | 3 | 1.05 | 10 | 3 |
| SRS | Autism | Male | 80.42 | 21.41 | 421 | 77 |
| | Autism | Female | 85.95 | 22.07 | 61 | 88 |
| | Control | Male | 38.43 | 15.25 | 337 | 41 |
| | Control | Female | 39.93 | 12.13 | 120 | 42 |

and has shown high overlap with manual quality control labelling[38]. The index counts the number of times the freesurfer has had to interpolate surface gaps during the reconstruction to ensure a continuous outcome surface. As such the index is effectively a measure for the reliability of the surface reconstruction and the resulting CT estimates. In the full sample we found a small but significant difference in both hemispheres (Supplementary Fig. 2) with the autism group having overall slightly worse scan quality ($d = 0.176$ for left and $d = 0.187$ for right hemispheres, respectively). Therefore, we chose to exclude the top 10% of subjects with an extreme Euler index (corresponding to a Euler index of ~300) and reran the Matchit genetic matching algorithm to check for matched samples. To further ensure adequate control for scan quality we included the index itself as a confound variable in all models.

Across both ABIDE I and ABIDE II CT was extracted for each subject using two different parcellations schemes: an approximately equally sized parcellation of 308 regions (~500 mm² each parcel)[33,56,57] and a parcellation of 360 regions derived from multi-modal features extracted from the Human Connectome Project (HCP) dataset[58]. The 308-region parcellation was constructed in the FreeSurfer fsaverage template by subdividing the 68 regions defined in the Desikan–Killiany atlas[59]. Thus, each of the 68 regions was sequentially sub-parcellated by a backtracking algorithm into regions of ~500 mm², resulting in a high-resolution parcellation that preserved the original anatomical boundaries defined in the original atlas[33]. Surface reconstructions of each individual were co-registered to the fsaverage subject. The inverse transformation was used to map both parcellation schemes into the native space of each participant.

**Statistics and reproducibility.** Because of power limitations in past work with small samples, we conducted an a priori statistical power analysis indicating that a minimum case-control effect size of $d = 0.1752$ could be detected at this sample size with 80% power at a conservative alpha set to 0.005[60]. For correlational analyses looking at brain–behaviour associations, we examined a subset of patients with the data from the SRS (N$_{autism\_male}$ = 421) and ADOS total scores (N$_{autism\_male}$ = 505). With the same power and alpha levels, the minimum effect for SRS is $r = 0.1765$ and $r = 0.1651$ for the ADOS.

There are likely many variables that contribute to variability in CT between individuals and across the brain. In order to visually assess the contribution of some prominent sources of variance we adopted a visualization framework derived from gene expression analysis (http://bioconductor.org/packages/variancePartition)[61] and included the most commonly available covariates in the ABIDE dataset: age, sex, diagnosis, scanner site, full-scale IQ, verbal IQ, handedness and SRS. Given that ABIDE was not designed as an integrated dataset from the outset, it seems plausible that the scanner site might be related to autism or autism-related variables (e.g., some sites might have different case-control ratios or only recruited specific subgroups). Figure 5 shows the ranked contribution of those covariates. Perhaps unsurprisingly, scanner site and age proved to be the most dominant sources of variance (each explaining on average around 15% of the total variance). Our initial conventional analysis was aimed to delineate potential broad case-control differences, as has been done in previous studies[14,27]. We used a linear mixed effects model with scanner site

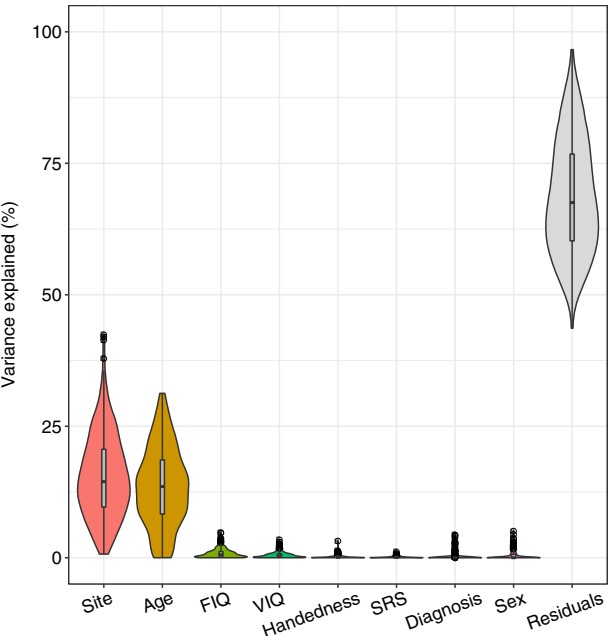

**Fig. 5 Explained variance in cortical thickness for each covariate.** Age age at the time of scanning, FIQ functional intelligent quotient, VIQ verbal intelligence quotient, SRS total score of the social responsive scale, Diagnosis diagnostic group, i.e. ASD or TD.

as a random effect. Given the potentially strong contribution of age we chose to include this as fixed effects covariates in the model. Multiple comparison correction was implemented with Benjamini–Hochberg FDR at $q < 0.05$[62]. All models also included Euler indices[38] and mean framewise displacement[37] as confound regressors (see also Supplementary Fig. 3 for sensitivity analyses on these confound regressors).

**Normative modelling reliability.** To assess the reliability of the normative w-score we permuted the normative sample (1000 bootstraps, with replacement) and computed 1000 permuted w-scores for each individual and each brain region. To subsequently quantify the reliability of the w-score we computed an FDR corrected analogous p-value for each subject by computing the absolute position of the real w-score in the distribution of permuted w-scores. The rationale being that if a real w-score would be in the top 5% of the bootstrapped distribution it would likely not be a reliable score (e.g. the score would be influenced by only a small subset of the normative data). The median number of brain regions per subject with a significant p-value in the normative sample was 1 (out of 308), indicating that the normative sample is topologically robust and that the w-score is a robust reflection of atypicality. More details on the bootstrapping procedure are provided in the supplementary material (Supplementary Fig. 4). To further assess the distribution in the normative group we also conducted one-sample linear mixed effects modelling in the normative group only to determine if any of all brain regions would show outlier consistency. There were no brain regions for which the w-score showed a deviation significant from zero in the normative group (even without correcting for multiple comparisons across all brain regions).

Because w-score maps are computed for each individual, we ran hypothesis tests at each brain region to identify regions that show on-average non-zero w-scores stratified by sex (FDR corrected at $q < 0.05$). To assess the effect of age-related individual outliers on the global case-control differences we re-ran the hypotheses tests on w-scores after removing region-wise individual outliers (based on a 2 SD cut-off). Although for clarity the present manuscripts present only results on CT, for completeness results from the same analysis on cortical volume, surface area and gyrification are shown in Supplementary Figs. 9–12.

Unfortunately, despite a significant female sub-group, the age-wise binning greatly reduced the number of bins with enough data-points in the female group. Given the reduced sample size in the female group and the known interaction between autism and biological sex[63,64], as well as the known sex differences in developmental trajectories[65], we conducted normative modelling on the male group only (Fig. 2a).

To explore isolated subsets of individuals with significant age-related CT atypicality, we used a cut-off score of 2 standard deviations (i.e. $w \geq 2$ or $w \leq 2$). This cut-off allows us to isolate specific ASD patients with abnormal CT relative to age-norms for each individual brain region. We then calculated sample prevalence (percentage of all ASD patients with atypical w-scores), in order to describe how frequent such individuals are in the ASD population and for each brain region individually. A sample prevalence map can then be computed to show the

frequency of these patients across each brain region. We also wanted to assess how many patients have markedly atypical w-scores (beyond 2 SD) across a majority of brain regions. This was achieved by computing an individual global w-score ratio as follows:

$$gW = \frac{\Sigma |w| > 2}{\Sigma |w| < 2}$$

We also computed global w-score ratios for positive and negative w regions separately.

**Exploratory analyses.** In addition to assessing the effect of normative outlier on conventional case-control analyses we also conducted some exploratory analysis on the normative w-scores. First, to explore whether the w-scores reflect a potentially meaningful phenotypic feature we also computed Spearman correlations for each brain region between the most commonly shared phenotypic features in ABIDE: ADOS, SRS, SCQ, AQ, FIQ and Age. Resulting p-values matrices were corrected for multiple comparisons using Benjamini–Hochberg FDR correction and only regions surviving and FDR-corrected p-value of < 0.05 are reported.

Finally, we explored whether the raw CT values could be used in a multivariate fashion to separate groups by diagnosis or illuminate stratification within ASD into subgroups. Here we used k-medoid clustering on t-distributed stochastic neighbour embedding (tSNE)[66]. Barnes–Hut tSNE was used to construct a two-dimensional embedding for all parcels in order to be able to run k-medoid clustering in a 2D representation and in order to visually assess the most likely scenario within the framework suggested by Marquand and colleagues[21]. Next, we performed partitioning around medoids (PAM), estimating the optimum number of clusters using the optimum average silhouette width[67]. Details of this exploratory analysis are reported in the supplementary materials (Supplementary Fig. 13).

**Reporting summary.** Further information on research design is available in the Nature Research Reporting Summary linked to this article.

## Data availability
All data is openly available on GitHub[32], this includes all measures extracted from the raw imaging data alongside the relevant phenotypic and quality control measures. Original unprocessed neuroimaging data is openly available through the ABIDE consortium: http://fcon_1000.projects.nitrc.org/indi/abide/abide_I.html.

## Code availability
All code is openly available on GitHub[32], Cohen's d were computed using: https://github.com/mvlombardo/utils/blob/master/cohens_d.R and the centiles cross-validation code can be found in https://github.com/deep-introspection/PyNM.

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

## Acknowledgements

This work was supported by a European Research Council (ERC) Starting Grant (755816; AUTISMS) awarded to M.V.L. R.R.-G. was funded by the Guarantors of Brain. J.S. was funded by the National Institutes of Health Oxford-Cambridge Scholars Program. R.A.I.B. was funded by a British Academy Post-Doctoral Fellowship and Autism Research Trust. Data were curated and analysed using a computational facility funded by an MRC research infrastructure award (MR/M009041/1) and supported by the NIHR Cambridge Biomedical Research Centre. The views expressed are those of the authors and not necessarily those of the NHS, the NIHR or the Department of Health and Social Care.

## Author contributions

R.A.I.B., R.R.-G., J.S. and M.V.L. designed the experiment. R.A.I.B., R.R.-G., J.S., S.T., G.D. and M.V.L. conceived and implemented all analyses. R.A.I.B., R.R.-G., J.S., S.T., G.D. and M.V.L. wrote the manuscript.

## Competing interests

The authors declare no competing interests.
