## [Peer Review File · Communications Biology]

Reviewers' comments:

Reviewer #1 (Remarks to the Author):

Bethlehem and colleagues applied normative modeling to characterize individualized metric of atypical cortical thickness in males ASD, based on ABIDE dataset. They identified that a large proportion of case-control differences in brain structures of ASD is driven by a subgroups of autistic individuals with highly age-atypical cortical thickness based on age-related norms. Overall, this is a well-conducted and well-written study with very interesting objectives and solid computation methodology. I've followed this work on biorxiv for a while, and am wondering why it has not been officially published in any peer-reviewed journal. I have few more points, which hopefully may improve soundness of the current findings.

1. I understand that authors have done QC steps, and put Euler index as a covariate in the model, Nonetheless, I still have concerns that the current results might be impacted by inter-individual variations in in-scanner motion, because individuals with extreme age-deviancy in atypical cortical thickness happened to have relatively poor image quality. This critical issue might be further alleviated by the following attempts: 1) test whether there is a correlations between Euler index with w-scores. 2) Compare Euler index between autistic individuals with higher age-normed deviancy and those within the age norms of CT. 3) further exclude additional 10% of participants with extreme Euler index from the remaining samples going through QC step. then see whether the major pattern still exist. If additional analyses based on either 1) or 2) yield significant findings, or the most remarkable pattern do not preserve after the practice of 3), then the main conclusions and inferences of the present findings should be considered problematic insofar as the in-scanner motion is taken into account.
2. Following the point 1: in the paragraph 3 of Discussion, the authors noted the limitation of a lack of phenotypic data of the subgroup with extreme age-deviancy, and seemed to explain that may be related to heterogeneous mechanisms of neurogenesis and development. Nonetheless, it was explicitly described in the Results that the T1 data of individuals with extreme deviancy in age-atypical cortical thickness showed relatively poor image quality. I think the likely influence of in-scanner motion on this result should be explicitly acknowledged and discussed here.
3. The authors have explained the rationale of using LOESS estimation for constructing the normative model, which I understand. I have concerns about the model fitting issue despite the strengths of LOESS estimation. The authors may want to provide the information to evaluate the goodness of fit to endorse the selection of LOESS estimation.

Reviewer #2 (Remarks to the Author):

The manuscript examines the possibility of using structural brain features (mainly cortical thickness but also cortical surface, volume as well as gyrification) in relation to age-related norms to try to better characterize ASD neural heterogeneity.

I am on the fence regarding this article. My general impression is that the idea is interesting, in principle, but results appear unconvincing to me. I may be convinced otherwise and remain open to a rebuttal, with supporting material/analyses, from authors.

Here are the main reasons for my ambivalence:

- 1) The authors identify outliers across all brain regions is 7.6% instead of the expected 4.55%. So, one could view this as meaning that the autism spectrum disorder group has a greater variance in cortical thickness than typically developing controls. If this is the case, that would indeed be an

interesting finding. However, it's not entirely clear to me that this is not simply driven by poor surface extraction due to subtle but systematic movement artifacts. See point 2.

2) It is not entirely clear to me that the Euler index does a good enough job at eliminating systematic subtle artifacts due to movement when comparing two groups with one known to perhaps moving more than the other. This is even more concerning here because the peak in outlier 'prevalence' happens to be children and that children are known to move more. Is it possible that FreeSurfer is particularly sensitive to movement artifacts in some regions more than in others? Can this have driven some of the results? To further mitigate problematic surface extraction effects, the authors included the Euler index as a covariate. While this is likely to help, it implicitly assumes a simple linear effect of the Euler index on cortical thickness. Even if the Euler index as a covariate had a simple linear association with thickness, it is not clear that other reconstruction issues would be dealt with. On the other hand, movement typically leads to artifactually thinner cortices rather than thicker ones, and here, after outlier removal and statistical thresholding, the autism outlier group appears to have some regions with a thicker cortex and no regions with a thinner cortex. This is partially reassuring and soothes my concern a little but not entirely. Now, I am not saying, of course, that no autism versus typical controls imaging studies should be conducted but that in the wake of so many false positive and unreproducible findings in the imaging literature, I feel that extra care is needed here. My QC concerns would be satisfied if authors implemented a more thorough QC by, perhaps, focusing on the regions that came out as statistically significant and by addressing a QC caveat in the discussion.

3) The analysis has been restricted, for the autism spectrum disorder (ASD) group, to those matched in IQ with typically developing individuals. This leads to a mean IQ of 106 in the ASD group which is higher than the general population mean and not at all representative of autism spectrum disorder where about 50% have IQs below 70 and only about 3% have IQs above 115. While I understand the need to match for IQ given frequently reported associations between IQ and cortical thickness, I fear that this restriction severely undermines the generalizability of finding to the general population of individuals with autism spectrum disorder. I suggest carefully and clearly addressing this in the discussion and perhaps even in changing the title to reflect this fact (perhaps referring to high level functioning individuals with ASD instead of simply autistic males).

Minor issues to consider:

I suggest avoiding the use of the term "autistic" (used in the title) as this is no longer used in the DSM-5 and suggest being consistent throughout the manuscript by using autism spectrum disorder or ASD.

Page 5: Authors refer to a "w-score", stating that it is analogous to a z-score. To me, it is exactly a z-score from the equation they provided. Why not call it a regional z-score?

Page 6: Table 1 has a group labeled as NT (I am assuming they are referring to the typically developing group for which they use TD for the rest of the manuscript. I suggest TD throughout instead of using NT at times.

Page 8: authors state that 7.6% is much higher than the expected 4.55% but provide no statistical test supporting this statement.

Page 9: Authors write: "There are other interesting attributes about this subset of brain regions. With regard to age, these patients were almost always in the age range of 6-20, and were much less prevalent beyond age 20 (S5)". Shouldn't this be supplementary figure S6 instead of S5?

Regarding S5: Authors write, in their legend, that there are 14 subjects for which the ratio score exceeds 0.5 yet the plots are not about these 14 subjects (this might be confusing to the reader).

Reviewer #3 (Remarks to the Author):

The authors present a technique for evaluating cortical thickness changes in individual ASD patients. The approach is in contrast to the current standard group-level cortical thickness analysis paradigm, which is limited to identifying differences between groups.

Specific comments:

1. In the introduction the authors note that biological sex is likely to modulate ASD-related neuroanatomical differences. What is the evidence for this claim?
2. Introduction: the authors discuss heterogenous findings in previous ASD studies, however they don't discuss the possibility that there may be effectively no morphometric differences between ASD individuals and healthy controls. Some negative studies have been published.
3. The author refers to brain regions passing FDR correction in the results section. I found this phrasing a little difficult to understand; FDR correction refers to a subset of statistical tests that are deemed statistically significant at a threshold modified for multiple comparisons. I assume the authors mean that some regions have statistically significant differences in cortical thickness between ASD and healthy controls? Consider clarifying. In a related point since the journal has the results section before the methods it's unclear what sort of spatial scale the authors are referring to when they talk about brain regions - is it CT averaged over cortical parcellations or are they vertex-wise cortical thickness estimates? It might be useful to modify the results section text to make it easier for the reader.
4. The authors note limitations with the Euler index for quantifying image quality. Pardoe et al Neuroimage 2016 "Motion and morphometry in clinical and nonclinical populations" demonstrated that in-scanner head motion, estimated using fMRI scans in ABIDE subjects, were correlated with cortical thickness estimates. It would be helpful to assess average head motion in the participants with abnormal age-related cortical thickness trajectories to make sure that the results aren't driven by in-scanner head motion.
5. Further to this, once the participants with abnormal CT for their age have been identified, there is relatively little further investigation of other factors that may explain why these participants had abnormal CT. This would strengthen the manuscript. Similarly it would be helpful if the authors provided the specific study IDs of the participants they identified with abnormal CT in the supplementary material.

Reply to reviewers

We first thank the reviewers for their thoughtful, encouraging, and constructive feedback. We fully agree that the paper would benefit from additional sensitivity analyses, as seems to also be the consensus across all three reviewers. Thus, in addition to providing a point-by-point reply to each point addressed we would like to first address the common remarks in relation to motion and data quality.

In the original manuscript we included the Euler index as a confounder in our models yet the reviewers all raise interesting concerns that it is of course still possible that there are residual data quality related issues in our downstream analysis. We indeed also made that remark in relation to the individuals with more extreme ratio scores, which also had somewhat poorer image quality in some cases. To assess and subsequently address these issues more formally we conducted a number of additional analyses.

1. We downloaded and processed the resting-state fMRI from all individuals present in ABIDE 1 and ABIDE 2 to obtain individual measures of in-scanner motion, specifically their mean framewise displacement. Framewise displacement was calculated for every EPI volume with the method described by Power and colleagues^{1,2} and then the mean was extracted for every individuals' scan session. Although fMRI and MPRAGE images were acquired during the same session, we cannot discard that participants' movements vary between acquisition sequences. Notwithstanding, it has been shown that framewise displacement is not only highly consistent across sessions but it is also strongly associated with MPRAGE image quality, suggesting that this metric could be also used as a proxy of motion-induced artifacts on structural images³.

For 35 individuals no resting-state data was available and a further 15 individuals had too poor quality of imaging to reliably assess in-scanner head-motion. In the overlapping sample we assessed case-control differences in mean framewise displacement, correlation between the computed ratio scores and confound variables and regional correlation between w-scores and both and Euler indices.

As can be expected there were indeed case-control effects of head-motion in the resting-state data (A), there was no significant correlation between the Euler indices and the absolute and negative w-ratio scores, but a small significant correlation with the positive ratio scores (B). In addition, we indeed find small regional (mostly negative) correlations with the w-score (ranging from -0.18 to 0.14) (C)⁴. We already included Euler indices as a confounder in any downstream analyses from here (e.g. one-sample test of w-scores, case-control differences in CT etc.). However, the reviewers are correct in noting that additional sensitivity analyses would help ensure

our findings were not driven by any confounding variables and we thus followed this assessment by testing these effects more formally.

Panel A shows the case-control difference in mean framewise displacement, indicating a significantly higher mean framewise displacement in the autism group $t(1125.68) = 5.07, p < .001$. Panel B shows the Pearson r correlations between age and the confound variables included in our models. Correlations not passing FDR correction of $p < .05$ are marked with a cross. Panel C shows the spatial correlation of each ROI with the three included confound regressors in our model, all three show small correlations ranging from $r = -.18$ to $r = .14$ and were thus included in all subsequent analyses.

2. First we performed 5-fold cross validation on both mean framewise displacement as well as on the Euler index. In both cases separately we systematically removed the top 5% percent of individuals (e.g. highest motion or highest Euler) up to 25% of removed data and recomputed the Cohen's d values across all regions for our initial case control analysis. We then computed the spatial correlation across regions for each fold. We find that for both motion as well as Euler the resulting Cohen's d maps were highly consistent (lowest $r = 0.7$ at 75% of individuals) and only decreased linearly with sample size (D). The new figure below shows the spatial correlation in the resulting Cohen's d maps. The upper triangle shows the validation for iterative exclusion of high motion individuals up to 25% (fold 5), the lower triangle shows the same for iterative exclusion of high Euler index individuals up to 25% (fold 5).

D | 5-fold Cross-validation

Panel D shows the spatial correlation between Cohen's D maps from analyses where subject with either high motion (upper triangle) or high Euler indices (lower triangle) were iteratively excluded. The fold refers to the cohorts of exclusion ranging from 1 = 5% exclusion to 5 = 25% excluded.

- Subsequently, we also re-ran all our analyses including motion as a confounder and again quantified the spatial correlation in resulting maps with the original model that did not include motion. We find that for both case-control differences as well as the one-sample test these correlations were close to 1 (E-F).

E | Correspondence of one-sample w-score after motion inclusion

F | Correspondence of group effects after motion inclusion

Panel E shows the correspondence in the one-sample model with and without motion included. Models show highly similar spatial topology ($r = 1.00$, $p < .001$, $BF = Inf$). Panel F shows the significant spatial correspondence for the between group linear mixed effects model ($r = 0.96$, $p < .001$, $BF = Inf$).

- Then, in order to quantify any residual effects of motion on the whole-brain ratio scores we ran correlation analysis on all three of the ratios with mean framewise displacement in our overlapping sample. Here we find that there are some small residual correlations between FD and ratio-scores (all $BF < 10$, max $r = 0.16$) (G-I). But no clear evidence that the individuals identified as having an extreme ratio (>0.5) were disproportionately overlapping with individuals that also exhibit higher in-scanner head motion.

Panel G shows the small relation between the absolute w-score and mean framewise displacement ($r = 0.15$, $p < .001$, $BF = -4.89$). Panels H and J shows the same for the positive ($r = 0.09$, $p < .05$, $BF = 0.13$) and negative ($r = 0.16$, $p < .001$, $BF = -6.67$) ratio's respectively.

- Although from these analysis it did not appear to be the case that the top motion individuals were also the individuals we classified as statistical outliers, we followed up this more formally. Specifically, we removed the top 5% of motion individuals and recomputed the ratio scores, assessed a residual correlation between motion and absolute ratios and recomputed the spatial prevalence (J-L). This resulted in two of the 14 original outlier subjects no longer being in the sample.

Panel J shows the residual correlation between motion and the absolute ratio after excluding the top 5% of motion subjects from the ASD sample ($r = 0.10$, $p < .05$, $BF = -0.43$). Panel K shows the absolute w-score

ratio with the dotted line indicating the cut-off of 0.5 after excluding the top 5% of motion individuals from the ASD sample. Panel L replicates the main figure 3 of this thresholded sample.

6. Similarly, we recomputed all ratios after excluding the top 5% of individuals with a high Euler index as well as the spatial prevalence (M-O). None of the original 14 subjects identified as outliers in the original analyses were in the top 5% and thus this did not affect the ratio scores.

Panel M shows the residual correlation between the absolute w-score ratio and the Euler index after thresholding the ASD sample at 5% of Euler scores ($r = -0.07$, $p = .08$, $BF = 1.33$). Panel N shows the absolute w-score in the thresholded sample with the dotted line indicating the 0.5 cut-off. Panel O replicated main figure 3 in this thresholded sample.

A comprehensive description is now included in the supplementary materials:

“To further assess the potential impact of data quality in the present study we processed the resting-state fMRI data to obtain estimates of in-scanner head-motion in the form of framewise displacement. We found that similar to the Euler index there were systematic group differences in in-scanner head motion (Figure S3.A), we also find that there were small significant correlations between in-scanner absolute, negative and positive w-score ratios ($r = 0.15$, $r = 0.16$ and $r = 0.09$ respectively, all $p < .05$, Figure S3.B). Then we assessed whether the extracted w-scores were spatially correlated with either Euler or head motion and found small (mostly negative) correlations ranging from $r = -0.18$ to $r = 0.14$ (Figure S3.C). Thus, we subsequently included both Euler index and framewise displacement as confound variables in all models.

Then, to systematically evaluate whether either motion or reconstruction quality impacted any of our outcome measures we conducted a cross-validation analysis by systematically excluding the top 5% of motion subject and top 5% of Euler subjects and assessed the spatial correlation in resulting Cohen’s D maps (Figure S3.D). Resulting maps were highly consistent, with the lowest

correlation ($r = 0.7$) between the sample with 95% and 75% of Euler subjects included, which is reflective of a decrease in sample size. We also assess the effect on the resulting one-sample assessment of the w -score between a model including motion as a confound and a model not including the motion confound (Figure S3.E). This showed near perfect consistency in spatial topology ($r = 1.00$, $p < .0001$). In similar fashion we assessed the case-control differences in a model with and without motion (Figure S3.F) and again find high consistency in spatial topology ($r = 0.96$, $p < .0001$).

To more specifically assess the relation between w -score ratios and head-motion we visualised their individuals correlations and observed that despite the small correlations there was little indication that high w -score ratio individuals (ratio $> .5$) were also the individuals with high motion (Figure S3G-I). To systematically assess their influence, we removed the top 5% of motion autism individuals. We then reassessed the correlation (Figure S3.J), the absolute w -score ratio (Figure S3.K) and the spatial prevalence (Figure S3.L) and found this exclusion did not impact our original results. In the same manner we excluded the top percentage of autism individuals with a high Euler index from the analysis (Figure S3M-O) and again found no impact on our original results.”

7. Finally, all analyses now include mean framewise displacement as a confound regressor. These comparisons against the original data are included in a new supplement (and given the resolution the whole figure has been made available online as a high-resolution figure). Including motion as a confound variable did reduce the effect enough in two regions for them to no longer pass FDR corrections, though the overall spatial pattern was highly similar. Interestingly, the inclusion of motion had a much stronger impact on the conventional case control analysis (e.g. where w -score outliers are still included) compared to the analysis excluding the small proportion of normative outliers. In the conventional model the number of regions passing FDR corrections dropped from 38 to 27 with the inclusion of motion, though again the overall pattern was highly similar. Previous figure 2 has been updated accordingly as has the results section describing the outcome of this analysis.

“Our first analysis examined conventional case-control differences using linear mixed effect modelling including site, sex, age, in-scanner head motion⁴ and Euler index⁵ as covariates. As expected from prior papers utilizing large-scale datasets for case-control analysis^{6,7}, a small

subset of regions (8.7%, 27/308 regions) pass FDR correction. Of these regions, most are of small effect size, with 26 of the detected 27 regions showing an effect less than 0.2 standard deviations of difference (Figure 2A).”

“However, removal of outlier patients now revealed only 14 significant regions instead of 27 regions with small case-control differences - a 1.9-fold decrease in the number of regions detected. Indeed, the majority of case-control differences identifying small on-average effects were primarily driven by this small subset of highly atypical patients (Figure 2B). These remaining 14 regions with small on-average effects were restricted to areas near the posterior cingulate cortex, temporo-parietal cortex and areas of visual cortex.”

A| Case-control linear mixed effects model

B| Case-control linear mixed effects model after outlier removal

Figure 2: Case control difference analysis with linear mixed effect model. Panel A shows effect sizes for regions passing FDR correction for linear mixed effect modelling of conventional case control difference analysis. Cohen's d' values represent ASD – Control, thus blue denotes ASD<Control and red denotes ASD>Control. Panel B shows effect sizes for regions passing FDR correction after outlier removal for the same linear mixed effect modelling of conventional case control difference analysis.

We also added an extra section to the results to briefly describe these sensitivity analyses.

“Sensitivity analyses on the effects of reconstruction quality using Euler index as well as residual effects of in-scanner head motion from the resting-state acquisition did not reveal a significant impact on thresholded case-control differences or w -score. Specifically, systematic exclusion of

top motion and Euler subjects resulted in highly spatially consistent effect size maps (all $r < 0.7$). Individuals identified as statistical outliers did not have disproportionately high motion or high Euler indices. For more details see methods and supplementary materials (Figure S3)."

We would again like to thank the reviewers for bringing this important factor to our attention and while our main results were unaffected by the inclusion of motion (e.g. spatial topology of effect, number of significant regions, spatial prevalence of atypicality and proportion of individuals considered statistical outliers) the conventional analysis clearly was and the more conservative approach is indeed to include motion as a confound variable. Thus, the manuscript has been updated throughout and the suggested additional sensitivity analyses have been added to the supplementary materials (in addition to a separate section on motion on the accompanying GitHub repository). Please find a point-by-point reply on other points raised by the reviewers below.

Reviewer #1

Bethlehem and colleagues applied normative modeling to characterize individualized metric of atypical cortical thickness in males ASD, based on ABIDE dataset. They identified that a large proportion of case-control differences in brain structures of ASD is driven by a subgroups of autistic individuals with highly age-atypical cortical thickness based on age-related norms. Overall, this is a well-conducted and well-written study with very interesting objectives and solid computation methodology. I've followed this work on biorxiv for a while, and am wondering why it has not been officially published in any peer-reviewed journal. I have few more points, which hopefully may improve soundness of the current findings. 1. I understand that authors have done QC steps, and put Euler index as a covariate in the model, Nonetheless, I still have concerns that the current results might be impacted by inter-individual variations in in-scanner motion, because individuals with extreme age-deviancy in atypical cortical thickness happened to have relatively poor image quality. This critical issue might be further alleviated by the following attempts: 1) test whether there is a correlations between Euler index with w-scores. 2) Compare Euler index between autistic individuals with higher age-normed deviancy and those within the age norms of CT. 3) further exclude additional 10% of participants with extreme Euler index from the remaining samples going through QC step. then see whether the major pattern still exist. If additional analyses based on either 1) or 2) yield significant findings, or the most remarkable pattern do not preserve after the practice of 3), then the main conclusions and inferences of the present findings should be considered problematic insofar as the in-scanner motion is taken into account.

We thank the reviewer for their positive comments and clear path to addressing some of the open concerns. As per the suggestions we have now conducted extensive additional analysis on the potential influence of the data quality using both cross-validation, analysis of residual

correlation as well as analysis based on exclusion of the more severely affected subjects. We are now even more confident that our original results were not affected by the data quality but have nonetheless chosen to update all results with the more stringent inclusion of motion in addition to Euler as a confounder. Rather than comparing the Euler index between two groups based on a particular cutoff we show that there is no meaningful residual correlation between Euler and the ratio score. We have updated the main manuscript to reflect these changes and included all sensitivity analyses in the supplementary materials. In addition, we have added to our discussion a more comprehensive acknowledgement of potential confounders.

“Finally, although in-scanner head motion is a well-known confounder in resting-state connectivity studies^{1,2} it has recently been shown that the same motion may also affect structural image quality and surface reconstruction, especially in clinical cohorts⁴. To address this issue in the present analysis we included mean framewise displacement in our models. We find that, while this severely impacted the conventional case control analysis (e.g. reducing the number of significant ROI’s from 38 to 27), it did not impact the outlier thresholded analysis to the same extent. To further assess the sensitivity of motion on the present approach we include sensitivity analyses based on systematic removal of high motion subjects and find that the spatial topology of effects was strongly conserved. Given the impact on the conventional analysis approach we strongly encourage future studies to consider motion as an important confounder.”

2. Following the point 1: in the paragraph 3 of Discussion, the authors noted the limitation of a lack of phenotypic data of the subgroup with extreme age-deviancy, and seemed to explain that may be related to heterogeneous mechanisms of neurogenesis and development. Nonetheless, it was explicitly described in the Results that the T1 data of individuals with extreme deviancy in age-atypical cortical thickness showed relatively poor image quality. I think the likely influence of in-scanner motion on this result should be explicitly acknowledged and discussed here.

As per the previous point (and see also at the top of this reply), we now included several motion sensitivity analyses and find that it did not meaningfully affect our results. However, the reviewers did raise a point that caution is warranted when interpreting imaging results without including motion as an important confounder and in addition to updating our analyses with this we also acknowledge the issue of motion more explicitly in the discussion.

“Finally, although in-scanner head motion is a well-known confounder in resting-state connectivity studies^{1,2} it has recently been shown that the same motion may also affect structural image quality and surface reconstruction, especially in clinical cohorts⁴. To address this issue in the present analysis we included mean framewise displacement in our models. We find that, while this

severely impacted the conventional case control analysis (e.g. reducing the number of significant ROI's from 38 to 27), it did not impact the outlier thresholded analysis to the same extent. To further assess the sensitivity of motion on the present approach we include sensitivity analyses based on systematic removal of high motion subjects and find that the spatial topology of effects was strongly conserved. Given the impact on the conventional analysis approach we strongly encourage future studies to consider motion as an important confounder."

3. The authors have explained the rationale of using LOESS estimation for constructing the normative model, which I understand. I have concerns about the model fitting issue despite the strengths of LOESS estimation. The authors may want to provide the information to evaluate the goodness of fit to endorse the selection of LOESS estimation.

As requested we now provide more information on how parameter optimisation for LOESS was conducted. Bootstrapping analysis of the w-score reliability was done around this optimisation and showed high between subject and between region consistency.

"We used a local polynomial regression fitting procedure (LOESS)^{8,9}, where the local width or smoothing kernel of the regression was determined by the model that provided the overall smallest sum of squared errors using hyperparameter optimisation across 5-100% of the full age range using Brent's method¹⁰ as implemented in the R optim function from the stats package. We also assessed consistency of our output using centiles scoring and consistency of the normative model using extensive bootstrapping, both showed high outcome consistency (see Methods and Supplementary Materials)."

Reviewer #2

The manuscript examines the possibility of using structural brain features (mainly cortical thickness but also cortical surface, volume as well as gyrification) in relation to age-related norms to try to better characterize ASD neural heterogeneity.

I am on the fence regarding this article. My general impression is that the idea is interesting, in principle, but results appear unconvincing to me. I may be convinced otherwise and remain open to a rebuttal, with supporting material/analyses, from authors.

Here are the main reasons for my ambivalence:

1) The authors identify outliers across all brain regions is 7.6% instead of the expected 4.55%. So, one could view this as meaning that the autism spectrum disorder group has a greater variance in cortical thickness than typically developing controls. If this is the case, that would indeed be an interesting finding. However, it's not entirely clear to me that this is not simply driven by poor surface extraction due to subtle but systematic movement artifacts. See point 2. 2) It is not entirely clear to me that the Euler index does a good enough job at eliminating systematic subtle artifacts due to movement when comparing two groups with one known to perhaps moving more than the other. This is even more concerning here because the peak in outlier 'prevalence' happens to be children and that children are known to move more. Is it possible that FreeSurfer is particularly sensitive to movement artifacts in some regions more than in others? Can this have driven some of the results? To further mitigate problematic surface extraction effects, the authors included the Euler index as a covariate. While this is likely to help, it implicitly assumes a simple linear effect of the Euler index on cortical thickness. Even if the Euler index as a covariate had a simple linear association with thickness, it is not clear that other reconstruction issues would be dealt with. On the other hand, movement typically leads to artifactually thinner cortices rather than thicker ones, and here, after outlier removal and statistical thresholding, the autism outlier group appears to have some regions with a thicker cortex and no regions with a thinner cortex. This is partially reassuring and soothes my concern a little but not entirely. Now, I am not saying, of course, that no autism versus typical controls imaging studies should be conducted but that in the wake of so many false positive and unreproducible findings in the imaging literature, I feel that extra care is needed here. My QC concerns would be satisfied if authors implemented a more thorough QC by, perhaps, focusing on the regions that came out as statistically significant and by addressing a QC caveat in the discussion.

We thank the reviewer for their comprehensive assessment of our work and fully agree that motion and its resulting artefact should really have been included in our analyses in the first place. The reviewers are likely correct in hypothesising that motion may affect FreeSurfer reconstruction in a spatially dependent manner. Since it is unclear what that spatial dependency might be or how to best estimate that from a structural image reconstruction we chose to not conduct a ROI analysis on this but rather include motion throughout all regions and all analyses. Thus, we thoroughly assessed the effect and association of motion to our results and, while we find that overall it did not change the outcome of the analysis, we agree that the validity of our findings is considerably strengthened by these additional analyses and by including it in our model. In addition, we now more forcefully acknowledge the potential of these confounders to influence morphological measurements.

“Finally, although in-scanner head motion is a well-known confounder in resting-state connectivity studies^{1,2} it has recently been shown that the same motion may also affect structural image quality

*and surface reconstruction, especially in clinical cohorts*⁴. To address this issue in the present analysis we included mean framewise displacement in our models. In this line, Savalia et al.³ recently showed that framewise displacement is a sensitive proxy of motion-related bias in structural images. We find that, while this severely impacted the conventional case control analysis (e.g. reducing the number of significant ROI's from 38 to 27), it did not impact the outlier thresholded analysis to the same extent. To further assess the sensitivity of motion on the present approach we include sensitivity analyses based on systematic removal of high motion subjects and find that the spatial topology of effects was strongly conserved. Given the impact on the conventional analysis approach we strongly encourage future studies to consider motion as an important confounder.”

3) The analysis has been restricted, for the autism spectrum disorder (ASD) group, to those matched in IQ with typically developing individuals. This leads to a mean IQ of 106 in the ASD group which is higher than the general population mean and not at all representative of autism spectrum disorder where about 50% have IQs below 70 and only about 3% have IQs above 115. While I understand the need to match for IQ given frequently reported associations between IQ and cortical thickness, I fear that this restriction severely undermines the generalizability of finding to the general population of individuals with autism spectrum disorder. I suggest carefully and clearly addressing this in the discussion and perhaps even in changing the title to reflect this fact (perhaps referring to high level functioning individuals with ASD instead of simply autistic males).

We thank the reviewer for pointing out this limitation to the generalizability of our approach and have taken the reviewers suggestion to list this as a caveat in our discussion. We chose not to change the title to include “high functioning ASD” as some recent literature has questioned whether higher IQ implies higher functioning (Tillmann et al. 2019). The exclusion of low IQ individuals and subsequent matching thus does not necessarily restrict the sample to what may normally be defined as high-functioning. We added the following caveat to our discussion to emphasize this.

“Fourth, our current sample was matched on IQ and as a result excluded individuals with low IQ scores (< 70). While higher IQ does not automatically imply higher overall functioning⁴⁸ it does limit the generalisability of our findings to individuals with normal to high IQ.”

Minor issues to consider:

I suggest avoiding the use of the term “autistic” (used in the title) as this is no longer used in the DSM-5 and suggest being consistent throughout the manuscript by using autism spectrum disorder or ASD.

This has been changed to ASD throughout the paper

Page 5: Authors refer to a “w-score”, stating that it is analogous to a z-score. To me, it is exactly a z-score from the equation they provided. Why not call it a regional z-score?

We agree that this is similar to a z-score yet have explicitly chosen not to refer to it as such since z-scores are more commonly associated with normalised scores within a population. Thus although calculation is the same we chose to use w-score to emphasise that the measure in the autism group is a z-score that is relative to the control group (not a z-score relative to the same group or to the combined group).

Page 6: Table 1 has a group labeled as NT (I am assuming they are referring to the typically developing group for which they use TD for the rest of the manuscript. I suggest TD throughout instead of using NT at times.

This table and text have been updated as per the reviewers suggestion.

Page 8: authors state that 7.6% is much higher than the expected 4.55% but provide to statistical test supporting this statement.

The chi-square test statistics are now included.

“This difference from an expected proportion of 5% in the present sample corresponds to a X^2 of 3.85 (with Yates continuity correction ¹¹) that is significant at $p < .05$.”

Page 9: Authors write: “There are other interesting attributes about this subset of brain regions. With regard to age, these patients were almost always in the age range of 6-20, and were much less prevalent beyond age 20 (S5)”. Shouldn’t this be supplementary figure S6 instead of S5?

With the addition of more supplemental analyses described above all figure indices have shifted and in the updated manuscript all references have been double-checked.

Regarding S5: Authors write, in their legend, that there are 14 subjects for which the ratio score exceeds 0.5 yet the plots are not about these 14 subjects (this might be confusing to the reader).

The caption has been updated.

Reviewer #3

The authors present a technique for evaluating cortical thickness changes in individual ASD patients. The approach is in contrast to the current standard group-level cortical thickness analysis paradigm, which is limited to identifying differences between groups.

Specific comments:

1. In the introduction the authors note that biological sex is likely to modulate ASD-related neuroanatomical differences. What is the evidence for this claim?

Multiple studies have shown sex*diagnosis interaction effects where the case-control effects present in one sex are statistically quite different than the same contrast in the other sex (e.g., Lai et al., 2013, Brain; Nordahl et al., 2011, 2015; Schaer et al., 2015; Beacher et al., 2012; Zeestraten et al., 2017, Transl Psychiatry). Furthermore, studies that stratify by sex and specifically examine CT find qualitative and quantitative distinctions between the sexes as well as sex-specific associations with symptom severity (Bedford et al., 2020). We have added a specific reference to this Bedford et al., paper to the statement about how biological sex modulates neuroanatomical differences with respect to CT.

2. Introduction: the authors discuss heterogenous findings in previous ASD studies, however they don't discuss the possibility that there may be effectively no morphometric differences between ASD individuals and healthy controls. Some negative studies have been published.

The reviewer raises a very interesting point that admittedly we had not acknowledged as explicitly as we probably should have. It is indeed possible that no morphometric differences exist and in fact our paper highlights that much of the previous literature likely over estimated true group mean differences. By focusing on a more individualised assessment we also find that it is only a small subset of individuals in which a broad atypical morphology is found and that these in fact drive most of the case-control difference. Thus it is likely fair to consider that on average no morphometric differences exist, yet that there may be a small subgroup of individuals within the autism group that do show some atypicality. We have rephrased our introduction and discussion to reflect this notion and included references on null findings.

“However, the vast neuroimaging literature is also inconsistent, with reports of hypo- or hyper-connectivity, cortical thinning versus increased grey or white matter, brain overgrowth, arrested growth, or even lack of morphological difference altogether etc. ¹²⁻²¹, leaving stunted progress towards understanding mechanisms driving cortical pathophysiology in ASD and translating neuroimaging into clinical utility.”

“Furthermore, conventional case-control analyses may obscure more subtle individual differences as they assume on-average group differences. This is especially important in light of previously reported null-findings ⁶.”

“Utilizing normative modelling as a way of identifying and removing CT-atypical outlier patients, we find here that most small case-control differences are driven by a small subgroup of patients with high CT-atypicality for their age, which indeed begs the question of the existence of on-average atypical cortical morphology in autism ⁶.”

3. The author refers to brain regions passing FDR correction in the results section. I found this phrasing a little difficult to understand; FDR correction refers to a subset of statistical tests that are deemed statistically significant at a threshold modified for multiple comparisons. I assume the authors mean that some regions have statistically significant differences in cortical thickness between ASD and healthy controls? Consider clarifying. In a related point since the journal has the results section before the methods it's unclear what sort of spatial scale the authors are referring to when they talk about brain regions - is it CT averaged over cortical parcellations or are they vertex-wise cortical thickness estimates? It might be useful to modify the results section text to make it easier for the reader.

We thank the reviewer for this helpful suggestion and have now restyled the manuscript to fit better with the ordering of NPG journals also allowing us to further clarify that all analysis were done on cortical parcellations.

“All analyses were done on cortical thickness averaged within 308 cortical regions ²².”

4. The authors note limitations with the Euler index for quantifying image quality. Pardoe et al Neuroimage 2016 "Motion and morphometry in clinical and nonclinical populations" demonstrated that in-scanner head motion, estimated using fMRI scans in ABIDE subjects, were correlated with cortical thickness estimates. It would be helpful to assess average head motion in the participants with abnormal age-related cortical thickness trajectories to make sure that the results aren't driven by in-scanner head motion.

We thank the reviewer for this helpful reference. We have now included motion in our model and in addition have conducted several sensitivity analyses to assess the impact of motion on our results. While we find that motion did not significantly impact our findings we now report all findings from the more conservative approach that does include motion and have updated the manuscript to more explicitly acknowledge the issue of motion.

“Finally, although in-scanner head motion is a well-known confounder in resting-state connectivity studies^{1,2} it has recently been shown that the same motion may also affect structural image quality and surface reconstruction, especially in clinical cohorts⁴. To address this issue in the present analysis we included mean framewise displacement in our models. We find that, while this severely impacted the conventional case control analysis (e.g. reducing the number of significant ROI’s from 38 to 27), it did not impact the outlier thresholded analysis to the same extent. To further assess the sensitivity of motion on the present approach we include sensitivity analyses based on systematic removal of high motion subjects and find that the spatial topology of effects was strongly conserved. Given the impact on the conventional analysis approach we strongly encourage future studies to consider motion as an important confounder.”

5. Further to this, once the participants with abnormal CT for their age have been identified, there is relatively little further investigation of other factors that may explain why these participants had abnormal CT. This would strengthen the manuscript. Similarly it would be helpful if the authors provided the specific study IDs of the participants they identified with abnormal CT in the supplementary material.

We fully agree that it would be interesting to see whether these individuals would show any particular atypicality in other domains but unfortunately little additional phenotypic information was available on the individuals identified as outliers. The full table of ratio scores has now been made available online including the anonymised subject ID’s.

References

1. Power, J. D., Barnes, K. A., Snyder, A. Z., Schlaggar, B. L. & Petersen, S. E. Spurious but systematic correlations in functional connectivity MRI networks arise from subject motion. *Neuroimage* **59**, 2142–2154 (2012).
2. Van Dijk, K. R. a., Sabuncu, M. R. & Buckner, R. L. The influence of head motion on intrinsic functional connectivity MRI. *Neuroimage* **59**, 431–438 (2012).
3. Savalia, N. K. *et al.* Motion-related artifacts in structural brain images revealed with independent estimates of in-scanner head motion. *Hum. Brain Mapp.* **38**, 472–492 (2017).
4. Pardoe, H. R., Kucharsky Hiess, R. & Kuzniecky, R. Motion and morphometry in clinical and nonclinical populations. *Neuroimage* **135**, 177–185 (2016).
5. Rosen, A. F. G. *et al.* Quantitative assessment of structural image quality. *Neuroimage* **169**, 407–418 (2018).
6. Haar, S., Berman, S., Behrmann, M. & Dinstein, I. Anatomical Abnormalities in Autism? *Cereb. Cortex* **26**, 1440–1452 (2016).
7. van Rooij, D. *et al.* Cortical and Subcortical Brain Morphometry Differences Between Patients With Autism Spectrum Disorder and Healthy Individuals Across the Lifespan: Results From the ENIGMA ASD Working Group. *Am. J. Psychiatry* **175**, appi.ajp.2017.1 (2018).
8. Cleveland, W. S., Devlin, S. J. & Grosse, E. Regression by local fitting: Methods, properties, and computational algorithms. *J. Econom.* **37**, 87–114 (1988).
9. Lefebvre, A. *et al.* Alpha Waves as a Neuromarker of Autism Spectrum Disorder: The Challenge of Reproducibility and Heterogeneity. *Front. Neurosci.* **12**, 662 (2018).
10. Brent, R. P. An algorithm with guaranteed convergence for finding a zero of a function. *Comput. J.* **14**, 422–425 (1971).
11. Yates, F. Contingency tables involving small numbers and the χ^2 test. *Supplement to the Journal of the Royal Statistical Society* **1**, 217–235 (1934).
12. Hong, S.-J., Bernhardt, B. C., Gill, R. S., Bernasconi, N. & Bernasconi, A. The spectrum of structural and functional network alterations in malformations of cortical development. *Brain* **140**, 2133–2143 (2017).
13. Hong, S.-J. *et al.* Atypical functional connectome hierarchy in autism. *Nat. Commun.* **10**, 1022 (2019).
14. Hong, S.-J., Valk, L., Martino, A. D., Milham, M. P. & Bernhardt, B. C. Multidimensional Neuroanatomical Subtyping of Autism Spectrum Disorder. 1–11 (2017).
15. Bedford, S. A. *et al.* Large-scale analyses of the relationship between sex, age and intelligence quotient heterogeneity and cortical morphometry in autism spectrum disorder. *Mol. Psychiatry* (2019) doi:10.1038/s41380-019-0420-6.
16. Schuetze, M. *et al.* Morphological Alterations in the Thalamus, Striatum, and Pallidum in Autism Spectrum Disorder. *Neuropsychopharmacology* **41**, 2627–2637 (2016).
17. Vissers, M. E., Cohen, M. X. & Geurts, H. M. Brain connectivity and high functioning autism: a promising path of research that needs refined models, methodological convergence, and stronger behavioral links. *Neurosci. Biobehav. Rev.* **36**, 604–625 (2012).
18. Ecker, C. *et al.* Brain surface anatomy in adults with autism: the relationship between surface area, cortical thickness, and autistic symptoms. *JAMA Psychiatry* **70**, 59–70 (2013).
19. Lai, M.-C. *et al.* Biological sex affects the neurobiology of autism. *Brain* **136**, 2799–2815 (2013).
20. Yang, D. Y.-J., Beam, D., Pelphrey, K. A., Abdullahi, S. & Jou, R. J. Cortical morphological markers in children with autism: a structural magnetic resonance imaging study of thickness, area, volume, and gyrification. *Mol. Autism* **7**, 11 (2016).
21. Mensen, V. T. *et al.* Development of cortical thickness and surface area in autism spectrum disorder. *NeuroImage: Clinical* **13**, 215–222 (2016).
22. Romero-garcia, R., Atienza, M., Clemmensen, L. H. & Cantero, J. L. Effects of network resolution on topological properties of human neocortex. *Neuroimage* **59**, 3522–3532 (2012).

REVIEWERS' COMMENTS:

Reviewer #1 (Remarks to the Author):

I appreciate that the authors have addressed my concerns. I'm happy to endorse the manuscript in current form.

Reviewer #2 (Remarks to the Author):

The authors have addressed all my comments appropriately. I am hence satisfied and feel that this paper should be published. Having said this, while I accept the authors' rationale for not using "high-functioning" in the title, I continue to think that the title is misleading because their work pertains to the very small proportion of individuals (probably less than 10%) with autism spectrum disorder that have IQs in the range of those in their study. I will let the editors decide if a change in title is necessary.

Reviewer #3 (Remarks to the Author):

The authors have addressed all my comments. Their extensive & thoughtful responses to the reviewer comments is appreciated.

Reply to reviewers

We first thank the reviewers and editor for their thoughtful, encouraging, and constructive feedback and their extensive help in improving the manuscript. We have changed the title to more accurately reflect that the current manuscript pertains to a subgroup of the broader autism population.